# Antimicrobial Compounds from Skin Secretions of Species That Belong to the Bufonidae Family

**DOI:** 10.3390/toxins15020145

**Published:** 2023-02-10

**Authors:** Rodrigo Ibarra-Vega, Alan Roberto Galván-Hernández, Hermenegildo Salazar-Monge, Rocio Zataraín-Palacios, Patricia Elizabeth García-Villalvazo, Diana Itzel Zavalza-Galvez, Laura Leticia Valdez-Velazquez, Juana María Jiménez-Vargas

**Affiliations:** 1Facultad de Ciencias Químicas, Universidad de Colima, Coquimatlán 28040, Mexico; 2Facultad de Medicina, Universidad de Colima, Coquimatlán 28040, Mexico; 3Consejo Nacional de Ciencia y Tecnología (CONACYT), Mexico City 03940, Mexico

**Keywords:** antimicrobial peptides, antifungal, antiviral, antiprotozoal, family Bufonidae, skin secretion

## Abstract

Skin secretions of toads are a complex mixture of molecules. The substances secreted comprise more than 80 different compounds that show diverse pharmacological activities. The compounds secreted through skin pores and parotid glands are of particular interest because they help toads to endure in habitats full of pathogenic microbes, i.e., bacteria, fungi, viruses, and protozoa, due to their content of components such as bufadienolides, alkaloids, and antimicrobial peptides. We carried out an extensive literature review of relevant articles published until November 2022 in ACS Publications, Google Scholar, PubMed, and ScienceDirect. It was centered on research addressing the biological characterization of the compounds identified in the species of genera *Atelopus*, *Bufo*, *Duttaphrynus*, *Melanophryniscus*, *Peltopryne*, *Phrynoidis*, *Rhaebo*, and *Rhinella*, with antibacterial, antifungal, antiviral, and antiparasitic activities; as well as studies performed with analogous compounds and skin secretions of toads that also showed these activities. This review shows that the compounds in the secretions of toads could be candidates for new drugs to treat infectious diseases or be used to develop new molecules with better properties from existing ones. Some compounds in this review showed activity against microorganisms of medical interest such as *Staphylococcus aureus*, *Escherichia coli*, *Bacillus subtilis*, Coronavirus varieties, HIV, *Trypanosoma cruzi*, *Leishmania chagasi*, *Plasmodium falciparum*, and against different kinds of fungi that affect plants of economic interest.

## 1. Introduction

The true toads are members of the family Bufonidae, which is formed by 52 genera and 644 species distributed worldwide [1,2]. These amphibians produce cutaneous secretions by two exocrine glands: mucous, and parotoid or granular. The first type is associated with respiration and hydration, while the parotoid glands secrete components related to defense, providing protection against predators and microorganisms [3,4,5,6]. These secretions consist of a complex composition of biomolecules containing alkaloids (dehydrobufotenine, atelopidtoxin, bufothionine, pumiliotoxin), biogenic amines (adrenaline, dopamine, serotonin), steroids (arenobufagin, Gamma-bufotalin, resibufogenin, telocinobufagin), peptides (alyteserin 1C, Bv8, cathelicidin), proteins, and others [7,8,9,10,11,12,13]. These components have various biological activities: antibacterial, anticancer, antifungal, anti-inflammatory, antiprotozoal, antitumoral, antiviral, insecticidal, and immunomodulatory [14,15,16,17,18,19,20,21]. In the data repository of antimicrobial peptides (DRAMP) it is reported that 1223 sequences were identified in amphibians. Still, few peptides have the potential to be characterized in clinical trials and to be used in humans as therapeutic agents [22]. Four antimicrobial peptides (AMPs) have been isolated from bufonid toads, but only one reached pre-clinical studies, as will be seen later. This review describes the diverse compounds of cutaneous secretions with antimicrobial activities identified in toads of the genera *Atelopus*, *Bufo*, *Duttaphrynus*, *Melanophryniscus*, *Peltopryne*, *Phrynoidis*, *Rhaebo*, and *Rhinella*, as well as of some chemically synthesized analogs that showed higher antimicrobial activity than the parent molecule dehydrobufotenine, and possible applications as antimicrobials. These compounds could be alternative drugs to control infections and antimicrobial resistance (AMR). With the outbreak of SARS-CoV2, 70% of the patients that presented with COVID-19 received antibiotics, which can lead to emergence of bacterial resistance [23,24,25]. Likewise, some molecules from toad skin secretions show antiviral activity. These compounds may be used against coronavirus variants that still are a worldwide problem, or can be used against some other viruses responsible for causing diseases [14,15,16,17,18]. Furthermore, compounds of secretions from toads may be used in agriculture as fungicides and pesticides [26,27]. Additionally, secretions from the parotoid glands of toads contain substances that have shown activity against intracellular parasites, so they can be an alternative for treating diseases caused by the genera *Plasmodium* spp., *Leishmania* spp., and *Trypanosome* spp. [11,28].

## 2. Methodology

### 2.1. Literature Review

All papers published until November 2022 on compounds from cutaneous toad secretions of the family Bufonidae with antimicrobial activities were considered for the literary review. The information search was carried out in the journals from ACS Publications, Google, Google Scholar, PubMed, and ScienceDirect, and the keywords used were “toad skin secretions”, “toad antimicrobials”, “antimicrobial”, “antibacterial”, “antiviral”, “antiparasitic”, and “antiprotozoal”.

### 2.2. Chemical Structure of Compounds and Determination of Molecular Weight

The determination of molecular weight, chemical formulas, and the chemical structure of the bufadienolides and alkaloids were performed with ChemBioDraw Ultra v12.0 software (PerkinElmer Inc., Waltham, MA, USA) [29]. In contrast, the secondary structure of the peptides was generated using the Maestro v.13.3.121 program (Schrodinger Inc., New York, NY, USA), and their molecular weight was determined with peptide mass calculator tools [30].

## 3. Antimicrobial Activities of Compounds from Toads’ Cutaneous Secretions

This review focuses on the bufadienolides, alkaloids, and antimicrobial peptides identified in toads’ cutaneous secretions with activity against bacteria, fungi, viruses, and parasites.

### 3.1. Compounds with Antibacterial Activity

After antibiotic misuse or overuse, it is almost inevitable that bacteria will become resistant to antibiotics. This kind of situation demands the development of new anti-infective drugs. The pharmaceutical industry attempts to meet this need through the timely redesign and modification of existing antibiotics and the development of new ones [31].

Research on the therapeutic potential of extracts and compounds isolated from true toads has intensified recently. Some studies support using secretions from bufonids in common treatments for bacterial diseases [32]. The water-soluble secretion of the parotid gland and the methanolic extract of the skin of the Sudanese toad *Duttaphrynus melanostictus* showed antibiotic activity against *Bacillus cereus*, *Escherichia coli*, *Klebsiella pneumoniae*, *Salmonella typhimurium*, *Staphylococcus aureus*, and *Staphylococcus epidermidis*; however, no action was found against methicillin-resistant *S. aureus* [33]. Sales et al. [34] observed that combining antibiotics of the aminoglycoside and β-lactam groups with the glandular secretion of *Rhinella jimi* led to a significant reduction in the minimal inhibitory concentration (MIC) value, mainly in strains of *E. coli*, *Pseudomonas aeruginosa*, and *S. aureus*. This indicated that components of the venom of *R. jimi* have significant pharmaceutical value in combination with available antibiotics. Other studies evaluated the antibiotic capacity of secretions from the dorsal skin of *Phrynoidis asper* extracted with deionized water, and demonstrated their action against the Gram-positive pathogens *S. aureus* and *Bacillus subtilis* with MIC of 12.25 ± 0.4 and 25 ± 1.3 μg/mL, respectively [35]. On the other hand, Thirupathi et al. [36] observed antibacterial activity from skin secretions from the Indian toad *Bufo melanostictus* on *E. coli*, *S. aureus*, *P. vulgaris*, and *K. pneumoniae* through the Kirby-Bauer Test.

In the Cundinamarca forest region of Colombia, *Rhinella* toads are used to treat erysipelas, an infection caused by the multi-drug-resistant bacterium *Streptococcus pyogenes* [37]. The *Rhinella marina* toad from the Brazilian Amazon has also been used to treat patients with erysipelas. Additionally, analysis of the crude extract and fractions of secretions from *R. marina* showed biological activity against *S. aureus* , *P. aeruginosa*, and *E. coli* [32]. In northeastern Brazil, the skin and fatty tissue of *Rhinella jimi* are used to treat some infections, such as, sore throat, flu, cough, and earache [13]. Bufadienolides are the most abundant compounds in the toads’ cutaneous secretions [38,39]. Their basic structure consists of a steroid nucleus of 17 carbon atoms arranged in four rings, and an α-pyrone ring at the C17-position, like in marinobufagin and telocinobufagin. It is important to mention that chemical modifications at various steroid molecule positions can generate remarkable differences in their antimicrobial activity (Figure 1). Cunha Filho et al. [40] used extraction with chloroform to isolate two bufadienolides, marinobufagin and telocinobufagin, from skin secretions of the Brazilian toad *Bufo rubescens*. The antibacterial capacity of both molecules was evaluated against *E. coli* ATCC 25922 and *S. aureus* ATCC 25213 and showed activities better than amoxicillin, imipenem, and trimethoprim antibiotics, especially against the Gram-positive bacteria (Table 1). Further, these authors determined that the two molecules can promote and increase cardiac contraction force. Both bufadienolides have been identified in toads of the genera *Atelopus* [41], *Rhinella*, *Rhaebo* [42], and *Peltopryne* [43].

The increasing spread of resistant bacteria has led to the constant search for bioactive compounds. The study of amphibian antimicrobial peptides (AMPs) has been employed as a folk remedy to treat bacterial infectious diseases. One group of these AMPs are cathelicidins, identified in the skin secretion of the Asian toad *Bufo bufo gargarizans*. Two sequences coding for these peptides were found through a skin transcriptomic analysis; they were chemically synthesized and named BG-CATH37 and BG-CATH(5-37). The first is formed by 37 amino acids, the second by 32, and both are rich in glycine and arginine residues. The antibacterial activity of these peptides was evaluated against *E. coli*, *S. aureus*, *P. aeruginosa*, and against aquatic bacteria (*Aeromorus hydrophia*, *Vibrio harveyi*, *Vibrio vulnificus*, *Vibrio splendidus*, *Lysogenic bacterium,* and *Streptococcus iniae*), these cathelicidins showed weak antimicrobial activity against human pathogenic bacteria (MIC > 200 μg/mL), but they showed strong antimicrobial activities against aquatic bacteria (MIC 3.125–40 μg/mL) from their habitat [44]. Other antibacterial peptides isolated from this toad were buforin I and buforin II (Table 1) [45]. These AMPs are from the histone family are composed of 39 and 21 amino acid residues, respectively, and can adopt a secondary structure of an alpha helix (Figure 2). The parent peptide buforin I was found to have a high degree of sequence similarity to the N-terminal segment of the histone H2A DNA-binding protein, leading to the assumption that this AMP is structurally similar to histone H2A and would have similar antibacterial effects to buforin II [46]. Buforin I showed antimicrobial activity against a wide range of microorganisms, including the Gram-positive, *Bacillus subtilis*, *S. aureus*, *Streptococcus mutans*, *Streptococcus pneumoniae*, *Pseudomonas putida*; the Gram-negative bacteria *E. coli*, *S. typhimurium*, *Serratia* sp., and the fungi *Candida albicans*, *Cryptococcus neoformans*, and *Saccharomyces cerevisiae* [45]. Subsequently, Roshanak et al. [47] demonstrated that this peptide has an antibacterial effect on the most resistant and sensitive microbial strains, *Staphylococcus salivarius* and *Clostridium perfringens*, with a minimum inhibitory concentration (MIC) of 4 and 16 μg/mL, respectively. Furthermore, buforin I inhibited the formation of biofilms. Additionally, the thermal stability of the peptide in plasma at different temperatures was evaluated, and hemolytic and cytotoxic assays were performed, indicating that this peptide has the pharmaceutical potential to be used as an antibiotic in infectious diseases (Table 1).

Meanwhile, buforin II, a peptide derived from the digestive system with endoproteinase Lys-C, retains the residues from Thr16 to Lys 36 of buforin I, and is more than 12 times more potent than buforin I [48]. A proline hinge region in buforin II is essential for inducing membrane permeation in *E. coli* [46,49]. It has been suggested that buforin II inhibits cellular processes by interfering with DNA and RNA metabolism [50].

### 3.2. Compounds with Antifungal Activity

Skin secretions of many toads contain peptides with antifungal activity. These peptides are stored in a granular gland located mainly in the skin of the dorsal region. Most amphibian species produce one or more peptides with potent activity against fungi [7].

The skin secretions of *Leptophryne cruentata* show antifungal activity against *Trichophyton mentagrophytes*; forty microliters caused an inhibition halo of 14.5 ± 2.9 mm diameter [51]. Furthermore, the skin secretions of *Rhinella icterica* showed antifungal activity against *Candida krusei* [28]. The bufadienolides arenobufagin, gamabufotalin, and telocinobufagin were identified in the skin secretions of the toad *Anaxyrus boreas*. These compounds inhibited the growth of the *Batrachochytrium dendrobatidis* strain, and their MIC were 12.9, 50.0, and 14.3 μM, respectively [7]. These bufadienolides were also identified in *Rhinella horribilis* [52]. The discovery of these compounds, effective against these fungal strains, will allow the development of therapeutic agents that treat infections caused in humans and other animals [53]. One alkaloid called dehydrobufotenine was found in the skin secretions of the toad *Bufo arenarum*. It showed fungicidal activity against phytopathogenic fungi that affect plants of economic interest. Further, forty-five analogs of dehydrobufotenine were synthesized and evaluated for their activity. Compounds **6**, **16c**, **16d**, **16h**, **16j**, and **19** showed similar or higher activity than the control carbendazim at 50 μg/mL. Dehydrobufotenine and most of its analogues showed higher activity against *Alternaria solani* (5–400%) than the controls chlorothalonil and carbendazim at 50 μg/mL. Their fungicidal activity is shown in Table 2.

### 3.3. Compounds with Antiviral Activity

Infectious diseases caused by viruses have impacted human life throughout evolutionary history and certainly in the adaptation of many civilizations. Molecular phylogenies support the hypothesis that some “modern” viruses, such as herpes or human papillomavirus (HPV), have co-evolved in association with some human ethnic groups and other vertebrate lineages and that these were dispersed around the world due to migration. The emergence or re-emergence of new diseases caused by viruses such as influenza, dengue, or hepatitis can be explained by the accelerated growth of populations and the density of human settlements, where the viruses were able to establish and persist, giving rise to the first epidemics [54].

In the last two years, the COVID-19 pandemic caused by the SARS-CoV-2 outbreak resulted in more than 50 million official cases and countless deaths. Due to this, many researchers focused on searching for alternatives to diagnose and treat this virus and others, such as influenza A (H1N1 and H5N1), HIV, Chikungunya, Zika, and Hepatitis C and B [55].

The importance of research on disease-causing viruses lies in the global public health problem and economic burden caused by the ability of viruses to transmit from nature to humans, triggering unpredictable outbreaks. The current strategy for managing viral diseases consists, on the one hand, of vaccines, whose development is usually slow, long, laborious, and costly; and on the other hand, of broad-spectrum antiviral treatments, an effective option to reduce human-to-human transmission [56]. For this reason, the bioactive components of toad skin secretions represent an important source for the development of new therapeutic agents that help in the treatment of various viral diseases.

Bufalin and cinobufagin are bufadienolides isolated from the toad skin secretion of the *Bufo bufo gargarizan*. Both are pharmacologically active components of huachansu or cinobufacini, a preparation of traditional Chinese medicine, which has been used as a treatment against hepatitis B virus (HBV). The activity of cinobufacini, bufalin, and cinobufagin was examined against HBV. The three of them were tested in the human cell line HepG2.2.15 at different concentrations, where all showed HBV secretion antigen inhibition activity at high concentrations (Table 3). Unlike bufalin and cinobufagin, the results demonstrated that cinobufacini exhibited higher activity against HBV at low concentrations. This was attributed to the specific inhibition of HBV messenger RNA expression [17,57,58].

In another study, bufalin and resibufogenin were studied for their effects against Epstein–Barr virus (EBV) or human herpesvirus 4, one of the most common viruses in humans. Both bufadienolides showed inhibitory activity on early activation of EBV antigen. In addition, bufalin showed in vivo inhibitory activity in two-stage carcinogenesis tests on mouse cutaneous papilloma caused by human papillomavirus (HPV) [59] and inhibition of rhinovirus [60].

Bufotenine is an alkaloid isolated from the secretions of *Rhinella jimi*. This compound was tested against rabies virus, which is responsible for a deadly zoonotic disease that affects more than 150 countries causing about 55,000 deaths annually. The study showed that bufotenine could inhibit the penetration of the rabies virus into mammalian cells through an apparent mechanism of competition for the nicotinic acetylcholine receptor. However, bufotenine is not presented as a proposal for drug development due to its hallucinogenic, cytotoxic, and psychotropic effects [61].

In 2021, Jin and coworkers tried to find broad-spectrum antiviral drugs to treat coronavirus infections. They assessed the antiviral activity of some cardiotonic steroids (bufadienolides) found in skin secretions of the toad species *Bufo bufo gargarizans*. The activity was tested against MERS-CoV, SARS-CoV, and SARS-CoV2 coronavirus varieties (Table 3). In the study, the authors identified antiviral activity. The IC_50_ values against coronavirus varieties were bufalin (0.016–0.544 μM), cinobufagin (0.017–0.616 μM), telocinobufagin (0.027–0.465 μM), bufotalin (0.027–1.630 μM), cinobufotalin (0.023–3.958 μM), and resinobufogenin (1.364–15.970 μM) (Table 3). Resinobufogenin showed low activity, and, sadly, bufalin and cinobufagin showed lethal activity in mice after intraperitoneal administration of 10 mg/kg/day in a 5-day repeated dose toxicity test. Cinobufotalin showed lethal activity in rats after 1mg/kg IV injection during pharmacokinetic studies. Telocinobufagin was the best candidate as a therapeutic drug for COVID-19 among the bufadienolides tested [16].

Further, the components of toad skin secretions can act against viruses that affect plants, as in the case of dehydrobufotenine (Figure 1). This alkaloid was found in skin secretions from *Bufo arenarum*. It was selected by Tian et al. [27] who designed and synthesized dehydrobufotenine analogs, with the goal of evaluating their biological activity. Of the forty-five synthesized analogs, compounds **12** and **17** showed ~50% higher antiviral activity against *Tobacco mosaic virus* (TMV) than dehydrobufotenine. Remarkably, the potency of these molecules was higher than ningnanmycin, which is perhaps the most effective anti-plant-virus agent (ningnanmycin inhibitory effect of 50–60% over TMV at 500 μg/mL). Compound **12** presented evidence of virus-assembly inhibition (Table 3) [27]. So, these two analogs could be considered new antiviral candidates.

### 3.4. Compounds with Antiparasitic Activity

A parasite is an “organism that lives on or in a host organism and gets its food from or at the expense of its” host [62]. The parasites use vectors such as mosquitos to reach their host. Some of these organisms cause diseases that have been associated with poverty [63] and tropical conditions [64]. Therefore warm weather in developing countries favors an increase in the case number, leading to deaths and other health problems [65].

Death due to parasitic diseases went down from 2000 to 2019. Still, in 2020, due to the SARS-CoV-2 pandemic, only malaria was determined to increase by 12%, estimating 627,000 deaths caused by the *Plasmodium vivax* parasite [64].

Chagas disease is caused by the parasite *Trypanosoma cruzi*, which is widespread in America (mainly rural zones of Latin America), affecting approximately 6–7 million people. The symptoms are nonspecific or asymptomatic in the acute phase of the infection. Patients suffer heart or digestive diseases only when the chronic stage is reached [28].

Drugs used in parasite-infection treatment are mainly formed by nitrogen rings, such as pyrimidine and imidazole, other substituents with nitrogen, and sulfhydryl groups are also common. There are antiparasitic drugs with broad spectra, such as albendazole and mebendazole; the majority are indicated against special cases, such as the use of mefloquine, which is a drug used against malaria produced by *P. falciparum*, a chloroquine-resistant strain, or suramin which is used on *Trypanosoma brucei rhodesiense* infection treatment [66].

For *T. cruzi*, benznidazole is a drug indicated for treating Chagas disease, especially in patients in the acute phase when the parasite is still in the blood. On the other hand, nifurtimox can be used in the acute phase or the chronic phase [67].

Toad venom comprises chemical compounds such as proteins, peptides, steroidal bufadienolides, alkaloids, and others [68]. The genus *Bufo* was divided into two genera in South America, *Rhinella* and *Rhaebo*. The secretions from nine different species (eight of *Rhinella* and one of *Rhaebo*) were analyzed biochemically, founding molecules in common. But, the authors realized that related species living in different habitats secrete distinct molecules, so they proposed that biological features influence skin secretion [68].

Telocinobufagin and hellebrigenin were isolated from *Rhinella jimi* and tested against *Leishmania chagasi* and *T. cruzi* (Table 4). Both bufadienolides inhibited the growth of the promastigotes from *Leishmania*, while hellebrigenin showed activity against trypomastigotes of *T. cruzi*. These molecules did not show cytotoxicity at 200 μg/mL, so that result is a promising for developing antiparasitic agents [28].

The skin secretion of the toads *R. marina* and *R. guttatus* were evaluated for their antiplasmodial activity against the *Plasmodium falciparum* W2 strain and showed a mortality above 70% against the parasite. Banfi et al. [69] isolated the compounds bufalin, dehydrobufotenin, marinobufotoxin, and marinobufagin of *R. marina* toads and evaluated their action in vitro on *P. falciparum* viability, showed values of IC_50_ of 1.329, 3.88, 5.31, and 1.557 μg/mL, respectively (Table 4). Although the bufalin was more active, only dehydrobufotenin demonstrated high selectivity for the parasites when analyzed in a cytotoxicity assay. Hence, it is a good candidate for the design of new therapeutic agents. On the other hand, secretions of *Rhinella marina* showed biological activity against *Leishmania guyanensis* and *Leishmania braziliensis* [32].

Meanwhile, ten bufadienolides isolated from *Rhinella alata* exhibited anti-trypanosomal activity, where the ranges of IC_50_ were between 1.329 and 14.343 μg/mL on *Trypanosoma cruzi* in an in vitro assay (Table 4). The anti-trypanosomal activity revealed that an arginyl-diacid at C-3 and an OH group at C-14 in the structure of bufadienolides were essential for the bioactivity observed.

Bufotalin and cinobufagin (IC_50_ 8.713 and 12.833 μg/mL against *T. cruzi*) presented low cytotoxicity (LD_50_ 20.446 and 23.897 μg/mL) in epithelial kidney monkey Vero cells (ATCC) [11].

## 4. Discussion

The skin secretions of the toads *Bufo bufo gargarizans* and *Bufo melanostictus* have been used in Chinese traditional medicine to treat bacterial infections, abdominal pain, and diarrhea [70]. These pharmacological properties are due to substances the secretions contain: bufadienolides, alkaloids, toxins, peptides, and proteins. In this work, we performed a review of the compounds with antimicrobial activity. The bufadienolides such as marinobufagin and telocinobufagin from *Bufo rubescens* presented lower antibacterial activity than the peptides buforins, especially against medically important bacteria such as *E. coli*, *S. aureus*, *S. pneumoniae*, and *S. typhimurium*. Buforin II showed better activity than buforin I. Although both peptides are rich in positive residues, the distribution of these in the alpha helix is crucial to induce bacterial killing [71]. In the case of buforin II, these positive residues are mainly distributed on the hydrophilic side. In this, the presence of arginine residues provides capabilities to interact with the negatively charged phospholipid headgroups of bacterial membranes. The studies with buforin II only included preclinical assays despite showing antibacterial activities similar to the derived peptide of magainin, called MSI-78, which was evaluated in Phase IIbAII clinical trials and demonstrated effectiveness in the treatment of impetigo, a skin infection caused by group A *Streptococcus* and *S. aureus* [72]. Buforins and cathelicidins could be used as parent peptides to design some analogs to increase their antimicrobial activity and prepare formulations of use to treat skin infections caused by bacteria.

Bufadienolides, also called cardiotonic steroids, contain a 6-membered lactone ring in the C17 position of the steroid nucleus (Figure 1) and have a broad spectrum of antimicrobial activities. Telocinobufagin and marinobufagin showed moderate antibiotic activity against *E. coli* (64 and 16 μg/mL) and low activity against *S. aureus* (128 μg/mL). Arenobufagin, gamabufotalin and marinobufagin showed medium antifungal activity (MIC 12.9, 50, 14.3 μg/mL, respectively). Dehydrobufotenine and some of its analogs showed better antifungal activity against *Alternaria solani* than the controls chlorothalonil and carbendazim.

The bufadienolides secreted from the skin of *Anaxyrus boreas* present an important opportunity for biology, since they present activity against the *Batrachochytrium dendrobatidis* strain, which affects the skin of many amphibians and has shown resistance to conventional treatment. This fungus has caused the extinction or near extinction of more than 200 species of amphibians, radically reprogramming ecosystems around the planet [7]. Alkaloids synthesized from the secretion of the toad *Bufo arenarum* showed potent fungicidal activity, surpassing fungicides such as chlorothalonil and carbendazim. Of all the synthesized components, it was found that compound **16h** was the main candidate to be a broad-spectrum fungicide [27].

On the other hand, telocinobufagin is one of the most interesting bufadienolides, it showed high activity against the MERS-CoV and SARS-CoV-2 coronavirus variants, and against *L. chagasi*, and, most importantly, it caused low cytotoxicity. Telocinobufagin is one of the crucial bufadienolides to consider a new medicine. In this review, we revised that slight differences in the molecules can cause significant changes in biological activities. There are still different compounds that have not been analyzed yet.

Steroidal compounds show high activity against parasites; 15 bufadienolides obtained from toad secretions demonstrated good activity (IC_50_ 126.2 to 1.32 μg/mL), but only 7 demonstrated viability in the therapeutic window. Commercial drugs used in treating parasites usually have hydrophobic areas that help in transportation across lipid barriers and contain amino hetero-substituted structures such as triazole and benzimidazole compounds. SAR-guided studies of aryl methylamino steroids showed steroid moieties as essential in biological activity to cross membrane barriers, and 3-OH and 17-OH substitution as essential to disable side effects of steroid hormones [73] like those observed in skin secretions of toads in this review.

## 5. Conclusions

In this review, we showed that skin secretions of toads are a source of biomolecules with diverse antimicrobial activities. In the secretions, bufadienolides are a diverse group of molecules. Some possess high antimicrobial activity but were reported to have toxic effects in mice, such as bufalin and cinobufagin. Telocinobufagin had moderate antifungal activity, high antiviral activity (against MERS-CoV, SARS-CoV, and SARS-CoV-2), and low toxicity. This molecule is a candidate for a new antiviral drug or could be a base for developing better antiviral agents.

Buforin II had high antibacterial activity killing Gram-negative and Gram-positive bacteria. This AMP is a clear candidate for a new antibiotic and could be used to treat skin and stomach infections. On the other hand, Buforin II could be used as a food preservative.

The study on dehydrobufotenine analogs was a clear example of designing new molecules with better properties; compounds **6**, **16c**, **16d**, **16h**, **16j**, and **19** increased their antifungal activity compared to dehydrobufotenine. They exhibited similar or higher activity than the commercial antifungals chlorothalonil and carbendazim. Furthermore, compounds **12** and **17** achieved higher activity that ningnanmycin, which may be the most effective anti-plant virus agent. Compounds **12** and **17** are therefore new antiviral candidates.

Up to date, 644 species in the Bufonidae family have been described, but the secretion of many of them has not been studied yet. We think there are still many antimicrobial compounds that can help us fight multiple diseases caused by microorganisms.

## Figures and Tables

**Figure 1 toxins-15-00145-f001:**
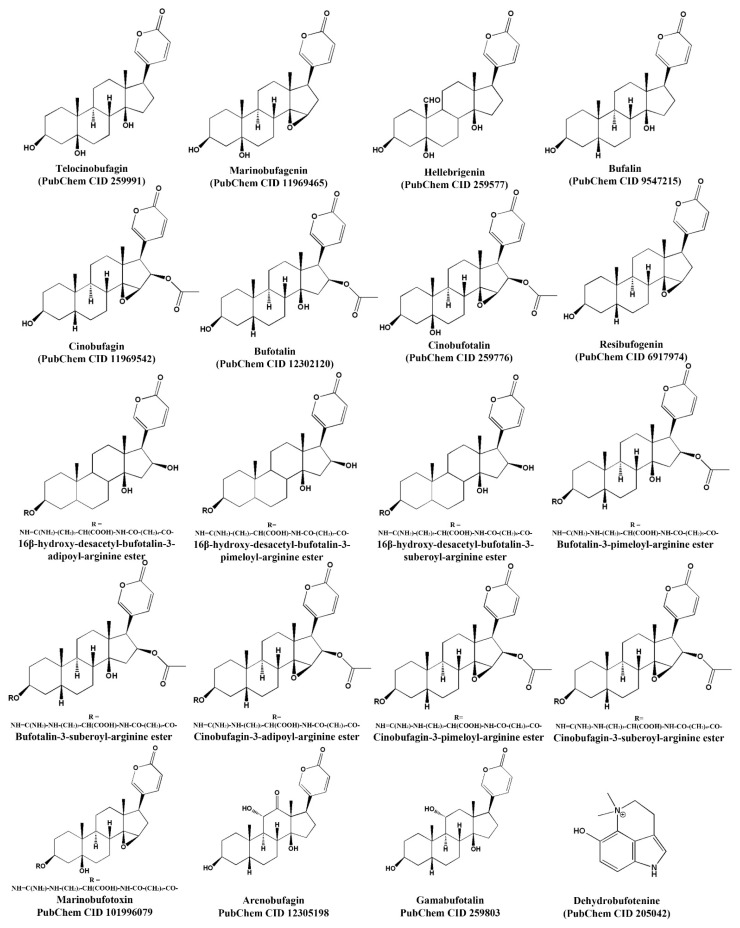
Chemical structure of molecules with antimicrobial activity found in toads’ skin secretions. The first 19 structures belong to the group of bufadienolides, and the last one is an alkaloid.

**Figure 2 toxins-15-00145-f002:**
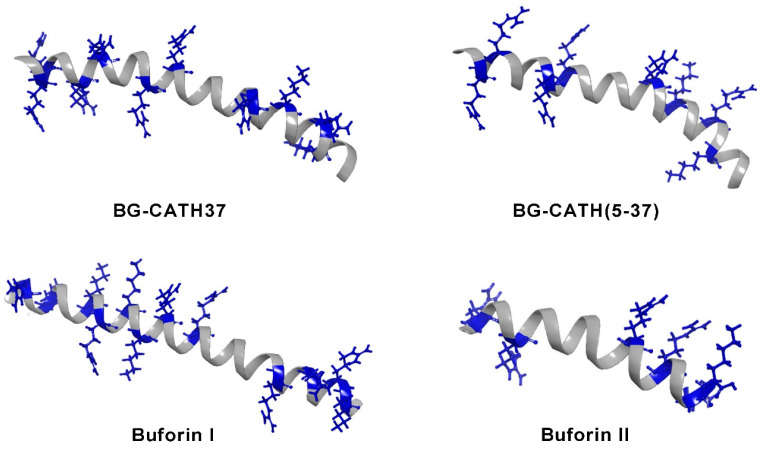
Secondary structure of antimicrobial peptides found in toads. These peptides adopt an alpha helical conformation. They are rich in lysine residues (represented in blue) and the first amino acids interacting with bacterial membranes. Component names are in bold.

**Table 1 toxins-15-00145-t001:** Compounds with antibacterial activity found in toads.

Species	Component	Chem. Form. M. Weight (Theo./Exp.)	Activity Against	Ref.
** *Bufo rubescens* **	Telocinobufagin (bufadienolide)	$C_{24}H_{34}O_{5}$ 402.52 Da/402.1609 Da	*E. coli* ATCC 25922 (64 μg/mL), *S. aureus* ATCC 25213 (128 μg/mL).	[40]
	Marinobufagin (bufadienolide)	$C_{24}H_{32}O_{5}$ 400.51 Da/400.1515 Da	*E. coli* ATCC 25922 (16 μg/mL), *S. aureus* ATCC 25213 (128 μg/mL).	[40]
** *Bufo bufo gargarizans* **	BG-CATH37 (peptide)	Amino acid sequence: SSRRPCRGRSCGPRLRGGYTLIGRPVKNQNRPKYMWV 4299.25 Da	*Streptococcus iniae* (12.5 μg/mL), *S. aureus* (>200 μg/mL), *Vibrio splendidus* (6.25 μg/mL), *Aeromonas hydrophila* (40 μg/mL), *P. aeruginosa* (200 μg/mL), *E. coli* (>200 μg/mL).	[44]
	BG-CATH(5-37) (peptide)	Amino acid sequence: PCRGRSCGPRLRGGYTLIGRPVKNQNRPKYMWV 3812.99 Da	*S. iniae* (12.5 μg/mL), *S. aureus* (>200 μg/mL), *V. splendidus* (3.125 μg/mL), *Aeromonas hydrophila* (12.5 μg/mL), *P. aeruginosa* (200 μg/mL), *E. coli* (>200 μg/mL).	[44]
	Buforin I (peptide)	Amino acid sequence: AGRGKQGGKVRAKAKTRSSRAGLQFPVGRVHRLLRKGNY 4363 Da/ MH^+^ 4309 Da	*Bacillus subtilis* (4 μg/mL), *S. aureus* (4 μg/mL), *S. pneumoniae* (4 μg/mL), *Streptococcus mutans* (8 μg/mL), *S. typhimurium* (4 μg/mL), *P. putida* (4 μg/mL), *E. coli* (8 μg/mL), *Serratia* spp. (8 μg/mL).	[45,47]
	Buforin II (peptide)	Amino acid sequence TRSSRAGLQFPVGRVHRLLRK 2434.9 Da/ MH^+^ 2432 Da	*S. aureus* (4 μg/mL), *S. pneumoniae* (4 μg/mL), *B. subtilis* (2 μg/mL), *S. mutans* (2 μg/mL), *P. putida* (2 μg/mL), *E. coli* (4 μg/mL), *Serratia* spp. (4 μg/mL), *S. typhimurium* (1 μg/mL).	[45]

Scientific names are shown in italics. Da, Daltons; MH^+^, molecular ion.

**Table 2 toxins-15-00145-t002:** Compounds with fungicidal activity from skin secretions of species that belong to the Bufonidae family, and dehydrobufotenine analogs.

Species	Molecule	Chem. Form. M. Weight	Activity Against	Ref.
** *Bufo arenarum* **	Dehydrobufotenine (alkaloid)	$C_{12}H_{15}N_{2}O^{+}$ 203.26 Da	*Alternaria solani* (50 μg/mL).	[27]
	Compound 6 (dehydrobufotenine analog)	$C_{16}H_{22}N_{2}O_{3}$ 290.36 Da	*Alternaria solani* (50 μg/mL), *Rhizoctonia solani* (50 μg/mL), *Botrytis cinereal* (50 μg/mL), *Cercospora arachidicola* (50 μg/mL).	
	Compound 16c (dehydrobufotenine analog)	$C_{22}H_{24}N_{4}O_{5}$ 424.45 Da	*Alternaria solani* (50 μg/mL), *Rhizoctonia solani* (50 μg/mL).	
	Compound 16d (dehydrobufotenine analog)	$C_{23}H_{26}N_{4}O_{6}$ 454.48 Da	*Sclerotinia sclerotiorum* (50 μg/mL). *Alternaria solani* (50 μg/mL), *Rhizoctonia solani* (50 μg/mL), *Botrytis cinereal* (50 μg/mL), *Cercospora arachidicola* (50 μg/mL).	
	Compound 16h (dehydrobufotenine analog)	$C_{20}H_{19}{\mathrm{BrN}}_{4}O_{4}$ 459.29 Da	*Sclerotinia sclerotiorum* (50 μg/mL), *Alternaria solani* (50 μg/mL), *Rhizoctonia solani* (50 μg/mL), *Botrytis cinereal* (50 μg/mL), *Fusarium graminearum* (50 μg/mL).	
	Compound 16j (dehydrobufotenine analog)	$C_{21}H_{19}F_{3}N_{4}O_{3}$ 432.40 Da	*Sclerotinia sclerotiorum* (50 μg/mL), *Alternaria solani* (50 μg/mL), *Botrytis cinereal* (50 μg/mL), *Cercospora arachidicola* (50 μg/mL).	
	Compound 19 (dehydrobufotenine analog)	$C_{19}H_{21}N_{2}O^{+}$ 293.38 Da	*Alternaria solani* (50 μg/mL), *Cercospora arachidicola* (50 μg/mL).	
** *Anaxyrus boreas* **	Arenobufagin (bufadienolide)	$C_{24}H_{32}O_{6}$ M. weight (theo./Exp.) 416.51 Da/416.9 Da	*B. dendrobatidis* (12.9 μg/mL).	[7]
	Gamabufotalin (bufadienolide)	$C_{24}H_{34}O_{5}$ M. weight (Theo./Exp.) 402.52 Da/403.4 Da	*B. dendrobatidis* (50 μg/mL).	
	Telocinobufagin (bufadienolide)	$C_{24}H_{34}O_{5}$ M. weight (Theo./Exp.) 402.52 Da/402.9 Da	*B. dendrobatidis* (14.3 μg/mL).	

Scientific names are shown in italics. Da, Daltons.

**Table 3 toxins-15-00145-t003:** Compounds with antiviral activity from skin secretions of species that belong to the Bufonidae family and dehydrobufotenine analogs.

Species	Molecule	Chem. Form. M. Weight	Activity	Ref.
** *Bufo arenarum* **	Dehydrobufotenine (alkaloid)	$C_{12}H_{15}N_{2}O^{+}$ 203.26 Da	At 500 μg/mL showed similar inactive, curative, and protective effects against TMV than the antiviral ribavirin.	[27]
	Compound **12** (dehydrobufotenine analog)	$C_{12}H_{16}N_{2}O$ 204.27 Da	At 500 μg/mL showed 5–7% higher inactivating, curative, and protective effects against TMV than the control ningnanmycin and 22–25% higher effects than ribavirin.	[27]
	Compound **17** (dehydrobufotenine analog)	$C_{18}H_{25}N_{2}O_{3}{}^{+}$ 317.40 Da	At 500 μg/mL showed 5–6% higher inactivating, curative, and protective effects against TMV than the control ningnanmycin and 20–21% higher effects than the control ribavirin.	[27]
** *Bufo rubescens* **	Telocinobufagin (bufadienolide)	$C_{24}H_{34}O_{5}$ 402.52 Da	IC_50_ 0.027 μM (anti-MERS-CoV ^a^), IC_50_ 0.465 μM (anti-MERS-CoV ^b^), IC_50_ 0.071 μM (anti-SARS-CoV ^a^), IC_50_ 0.142 μM (anti-SARS-CoV2 ^a^).	[16]
** *Bufo bufo gargarizans* **	Bufalin (bufadienolide)	$C_{24}H_{34}O_{4}$ 386.52 Da	IC_50_ 0.018 μM (anti- MERS-CoV ^a^), IC_50_ 0.544 μM (anti-MERS-CoV ^b^), IC_50_ 0.017 μM (Anti-SARS-CoV ^a^), IC_50_ 0.019 μM (Anti-SARS-CoV2 ^a^), IC_90_ 15 nM (anti-HIV-1), 100% lethal activity at 10 mg/kg (in mice, IP). 20 nM inhibits the secretion of 14.42% of HBsAg and 30.95% of HBeAg^c^ before three days of treatment, while at six days it inhibits the secretion of 11.36% of HBeAg ^c^ and 19.58% at 0.2 nM [16,17,18].	[16,17,18]
	Cinobufagin (bufadienolide)	$C_{26}H_{34}O_{6}$ 442.54 Da	IC_50_ 0.017 μM (anti- MERS-CoV ^a^), IC_50_ 0.616 μM (anti-MERS-CoV ^b^), IC_50_ 0.061 μM (Anti-SARS-CoV ^a^), IC_50_ 0.072 μM (Anti-SARS-CoV2 ^a^), IC_50_ 10.94 nM (anti-enterovirus 71), IC_90_ 40 nM (anti-HIV-1), 100% lethal activity at 10 mg/kg/day in a 2-day repeated dosage (in mice, IP). 1 nM inhibits the secretion of 8.28% of HBeAg ^c^ and 3.01% of HBcrAg ^d^ before three days of treatment, while at six days it inhibits the secretion of 7.01% of HBeAg ^c^ and 7.16% of HBcrAg ^d^.	[14,16,17,18]
	Bufotalin (bufadienolide)	$C_{26}H_{36}O_{6}$ 444.56 Da	IC_50_ 0.063 μM (anti- MERS-CoV ^a^), IC_50_ 1.630 μM (anti-MERS-CoV ^b^), IC_50_ 0.027 μM (Anti-SARS-CoV ^a^), IC_50_ 0.073 μM (Anti-SARS-CoV2 ^a^).	[16]
	Cinobufotalin (bufadienolide)	$C_{26}H_{34}O_{7}$ 458.54 Da	IC_50_ 0.231 μM (anti- MERS-CoV ^a^), IC_50_ 3.958 μM (anti-MERS-CoV ^b^), IC_50_ 0.429 μM (Anti-SARS-CoV ^a^), IC_50_ 0.399 μM (Anti-SARS-CoV2 ^a^), 100% lethal activity in rats at 1 mg/kg (IV).	[16]
	Resibufogenin (bufadienolide)	$C_{24}H_{32}O_{4}$ 384.51 Da	IC_50_ 1.612 μM (anti- MERS-CoV ^a^), IC_50_ 15.97 μM (anti-MERS-CoV ^b^), IC_50_ 1.364 μM (Anti-SARS-CoV ^a^), IC_50_ 1.606 μM (Anti-SARS-CoV2 ^a^), IC_50_ 218 nM (anti-enterovirus 71).	[14,16]

TMV, tobacco mosaic virus; ^a^ coronavirus in Vero cells (ATCC^®^ CCL-81™); ^b^ coronavirus in Calu-3 (ATCC^®^ HTB-55™) cells; ^c^ HBeAg, hepatitis B, e antigen; ^d^ HBcrAg, hepatitis B, core-related antigen; Da, Daltons; ${\mathrm{IC}}_{50},$ 50% inhibitory concentration; IV, intravenous injection, IP, intraperitoneal injection. Scientific names are in italics.

**Table 4 toxins-15-00145-t004:** Compounds with antiparasitic activity from skin secretions of species that belong to the Bufonidae family.

Species	Molecule	Chem. Form.M. Weight	Activity	Ref.
** *Rhinella jimi* **	Telocinobufagin (bufadienolide)	$C_{24}H_{34}O_{5}$ 402.52 Da	IC_50_ 61.2 μg/mL (*L. chagasi* promastigotes). 150 μg/mL reduced 37.0% of amastigotes in macrophages. At 200 μg/mL, cytotoxicity was not observed.	[28]
	Hellebrigenin (bufadienolide)	$C_{24}H_{32}O_{6}$ 416.51 Da	IC_50_ 126.2 μg/mL (*L. chagasi* promastigotes). 150 μg/mL reduced 21.5% of amastigotes in macrophages. IC_50_ 91.75 μg/mL (*T. cruzi* trypomastigotes) At 200 μg/mL, cytotoxicity was not observed.	
** *Rhinella alata* **	Bufotalin (bufadienolide)	$C_{26}H_{36}O_{6}$ 444.57 Da	IC_50_ 8.713 μg/mL (*T. cruzi* trypomastigotes), CC_50_ 20.46 μg/mL, Vero cells.	[11]
	16β-hydroxy-desacetyl-bufotalin-3-pimeloyl-arginine ester (bufadienolide)	$C_{37}H_{56}N_{4}O_{9}$ 700.4 Da	IC_50_ 6.093 μg/mL (*T. cruzi* trypomastigotes), CC_50_ 0.168 μg/mL, Vero cells.	
	Bufotalin-3-pimeloyl-arginine ester (bufadienolide)	$C_{37}H_{56}N_{4}O_{9}$ 742.91 Da	IC_50_ 3.41 μg/mL (*T. cruzi* trypomastigotes), CC_50_ 0.126 μg/mL, Vero cells.	
	16β-hydroxy-desacetyl-bufotalin-3-suberoyl-arginine ester (bufadienolide)	$C_{38}H_{58}N_{4}O_{9}$ 714.9 Da	IC_50_ 4.932 μg/mL (*T. cruzi* trypomastigotes), CC_50_ 0.579 μg/mL, Vero cells.	
	Bufotalin-3-suberoyl-arginine ester (bufadienolide)	756.94 Da C_40_H_60_N_4O_10	IC_50_ 3.079 μg/mL (*T. cruzi* trypomastigotes), CC_50_ 0.267 μg/mL, Vero cells.	
	Cinobufagin-3-adipoyl-arginine ester (bufadienolide)	$C_{38}H_{54}N_{4}O_{10}$ 726.87 Da	IC_50_ 12.138 μg/mL (*T. cruzi* trypomastigotes), CC_50_ 2.616 μg/mL, Vero cells.	
	Cinobufagin-3-pimeloyl-arginine ester (bufadienolide)	$C_{39}H_{56}N_{4}O_{10}$ 740.9 Da	IC_50_ 10.891 μg/mL (*T. cruzi* trypomastigotes), CC_50_ 0.533 μg/mL, Vero cells.	
	Cinobufagin-3-suberoyl-arginine ester (bufadienolide)	$C_{40}H_{58}N_{4}O_{10}$ 754.92 Da	IC_50_ 14.343 μg/mL (*T. cruzi* trypomastigotes), CC_50_ 0.188 μg/mL, Vero cells.	
	Cinobufagin (bufadienolide)	442.55 Da C_26_H_34_O_6_	IC_50_ 12.833 μg/mL (*T. cruzi* trypomastigotes), CC_50_ 23.897 μg/mL, Vero cells.	
** *Rhinella marina* **	Marinobufotoxin (bufadienolide)	$C_{38}H_{56}N_{4}O_{9}$ 712.89 Da	IC_50_ 3.778 μg/mL (*P. falciparum* W2), LD_50_ 6.337 μg/mL, WI-26VA4 cells.	[69]
	Bufalin (bufadienolide)	$C_{24}H_{34}O_{4}$ 386.52 Da	IC_50_ 1.329 μg/mL (*P. falciparum* W2), LD_50_ 3.436 μg/mL, WI-26VA4 cells.	
	Dehydrobufotenine (Alkaloid)	$C_{12}H_{15}{\mathrm{ON}}_{2}{}^{+}$ 203.26 Da	IC_50_ 3.884 μg/mL (*P. falciparum* W2), LD_50_ 47.92 μg/mL, WI-26VA4 cells.	
	Marinobufagin (bufadienolide)	$C_{24}H_{32}O_{5}$ 400.51 Da	IC_50_ 1.557 μg/mL (*P. falciparum* W2), LD_50_ 1.217 μg/mL, WI-26VA4 cells.	

Da, Daltons; ${\mathrm{IC}}_{50},$ 50% inhibitory concentration; CC_50_, 50% cytotoxic concentration; WI-26VA4, human pulmonary fibroblast cells. Scientific names are in italics.

## Data Availability

Not applicable.

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
