# Peer review of "Antimicrobial Compounds from Skin Secretions of Species That Belong to the Bufonidae Family"

_toxins, 2023, doi:10.3390/toxins15020145_

Round 1

Reviewer 1 Report

Thank you for the opportunity to review the paper entitled Antimicrobial compounds from skin secretions of species that belong to the Bufonidae family. I find this paper very interesting and valuable. The used literature is appropriate and the manuscript is well-structured. Nevertheless, I find some points that need revision:

First of all, the information of authors and their affiliation is completely missing.

All figures need to be in better quality

When the full name of microorganisms is written there should be space after the ATCC and the number.

the Discussion part as well as the conslusion part sholud be extended. For example with pros and cons of using skin secretion of toads as source of biomolecules. What is already known and what should be further investigated? 

Author Response

Reviewer #1: Thank you for the opportunity to review the paper entitled “Antimicrobial compounds from skin secretions of species that belong to the Bufonidae family”. I find this paper very interesting and valuable. The used literature is appropriate, and the manuscript is well-structured. Nevertheless, I find some points that need revision:

Answer: We appreciate your comments.

First of all, the information of authors and their affiliation is completely missing. All figures need to be in better quality.

Answer: Line 4-8 We have added the complete information of authors and their affiliations, and the quality and clarity of the figures were improved.

When the full name of microorganisms is written there should be space after the ATCC and the number.

Answer: We apologize, the change was performed in the Line 113

The Discussion part as well as the conclusion part should be extended. For example with pros and cons of using skin secretion of toads as source of biomolecules. What is already known, and what should be further investigated?

Answer: Thank you for your observations. We agree with you and changed the discussion and conclusion.

Reviewer 2 Report

The review is written in a style based on the principle of collecting articles and summarizing information from their abstracts. The author's processing of the available material (articles) can be traced weakly. It is not clear why this review is needed, what purpose the authors pursued. If the authors would like to tell that amphibians are a source of new substances for medicine, then they should start the introduction by mentioning how many structures of antimicrobial peptides of amphibian origin are known and deposited in databases (DBAASPAPD3, CAMPr4). What are the advantages and limitations of amphibian antimicrobials compared to other sources? Which ones are already in pre-clinical/clinical studies, and which ones have failed trials?

Lines 95-98 and Figure 1.3. I think that it is not necessary to devote much space to describing substances that are analogues of natural substances, but obtained using drug design and chemical synthesis.

 Table 1 BG-CATH37 is a peptide, as is Buforin I. Column "Activity": achieve uniformity in the description of the properties. I suggest: “active against E. coli ATCC 25922 (64 μg/mL), S. aureus ATCC 25213 (128 μg/mL)” and in the same way on the remaining lines. Note that such writing ">200 μg/mL" means that the substance is inactive against strains. Cardiac activity is not related to antimicrobial properties.

Lines 33-35. The sentence is incorrectly composed, rewrite it.  Compounds cannot be an alternative to some problem.

Lines 72-74. The full names of taxa should be written at their first mention in the text.

Lines 95-98 and Figure 1. Only those molecules that have antimicrobial properties need be mentioned here.

Lines 98-99. Delete this sentence

Line 103. What commercial antibiotics were mentioned?

Lines 116-117 Abbreviated names of taxa should be used when they are mentioned again in the text.

Lines 307-312. Difficult to understand the point. The sentence needs to be revised.

Author Response

Reviewer #2

The review is written in a style based on the principle of collecting articles and summarizing information from their abstracts. The author's processing of the available material (articles) can be traced weakly. It is not clear why this review is needed, what purpose the authors pursued. If the authors would like to tell that amphibians are a source of new substances for medicine, then they should start the introduction by mentioning how many structures of antimicrobial peptides of amphibian origin are known and deposited in databases (DBAASP, APD3, CAMPr4). What are the advantages and limitations of amphibian antimicrobials compared to other sources? Which ones are already in pre-clinical/clinical studies, and which ones have failed trials?

Answer: We thank this advice. We have modified the sections of the Abstract (lines 20-26) and introduction (lines 43-47). The preclinical studies were added in the discussion section (lines 348-354).

Lines 95-98 and Figure 1.3. I think that it is not necessary to devote much space to describing substances that are analogues of natural substances but obtained using drug design and chemical synthesis.

Answer: You are right; we have deleted the analogs of the text and Figures 1 and 2.

 Table 1 BG-CATH37 is a peptide, as is Buforin I. Column "Activity": achieve uniformity in the description of the properties. I suggest: “active against E. coli ATCC 25922 (64 μg/mL), S. aureus ATCC 25213 (128 μg/mL)” and in the same way on the remaining lines. Note that such writing ">200 μg/mL" means that the substance is inactive against strains. Cardiac activity is not related to antimicrobial properties.

Answer: We agree with this observation. We made the changes in all tables.

Lines 33-35. The sentence is incorrectly composed, rewrite it.  Compounds cannot be an alternative to some problem.

Answer: Thank you for your observation. The text was corrected (lines 53-54).

Lines 72-74. The full names of taxa should be written at their first mention in the text.

Answer: Thank you for your observation. The text was corrected (lines 93-94).

Lines 95-98 and Figure 1. Only those molecules that have antimicrobial properties need be mentioned here.

Answer: You are right; we have deleted the analogs of the text, and Figure 1

Lines 98-99. Delete this sentence.

Answer: We have deleted the sentence

Line 103. What commercial antibiotics were mentioned?

Answer: Thank you for your observation. The text was corrected (line 125).

Lines 116-117 Abbreviated names of taxa should be used when they are mentioned again in the text.

Answer: Thank you for your observation. The text was corrected (lines 139, 140, 152, 153).

Lines 307-312. Difficult to understand the point. The sentence needs to be revised.

Answer: We thank this advice. We have modified the sentence (lines 328-336).

Thank you very much for all your comments and suggestions. They were all addressed in the new manuscript. All changes were highlighted in yellow in the manuscript.

Round 2

Reviewer 1 Report

The authors have significantly improved the quality of the initial manuscript. Therefore, I recommend it for the publication in current form.